# Effect of Silicon Carbide Nanoparticles on the Friction-Wear Properties of Copper-Based Friction Discs

**DOI:** 10.3390/ma15020587

**Published:** 2022-01-13

**Authors:** Changsong Zheng, Zhiwei Ma, Liang Yu, Xu Wang, Liangjie Zheng, Li’an Zhu

**Affiliations:** 1School of Mechanical Engineering, Beijing Institute of Technology, Beijing 100081, China; zhengchangsong@bit.edu.cn (C.Z.); 3220200292@bit.edu.cn (Z.M.); 3220190331@bit.edu.cn (X.W.); 3120170250@bit.edu.cn (L.Z.); 2Jianglu Electromechanical Group Co., Ltd., Xiangtan 411100, China; 3220200280@bit.edu.cn

**Keywords:** wet clutch, copper-based friction materials, silicon carbide nanoparticles, friction-wear properties, microscopic morphology

## Abstract

To study the influence of nano-additives on the friction-wear characteristics of friction materials, the nano-sized silicon carbide particles which have excellent chemical and physical properties are considered to add in composite to form the modified friction material. The influence of the silicon carbide nanoparticles (SCN) on the friction-wear characteristics of copper-based friction materials (CBFM) is investigated via the SAE#2 (made in Hangzhou, China) clutch bench test with the applied pressure, rotating speed, and automatic transmission fluid (ATF) temperature taken into account. Moreover, the variations of friction torque and temperature are considered to evaluate the friction performance, and the variable coefficient is employed to describe the friction stability. The wear characteristics of friction materials are investigated by the disc changes in thickness and micro-morphology. The results show that the CBFM with SCN can provide a higher friction torque, which increased by 30% to 50% compared with CBFM. The variable coefficient of CBFM with SCN changes from 674 to 52 with the rotating speed raised from 600 rpm to 3000 rpm, which shows that the friction stability is relatively worse. Furthermore, the micromorphology shows that the CBFM with SCN has lower porosity and surface roughness, which increases the microscopic contact area and the coefficient of friction (COF). Simultaneously, the reduction in porosity also leads to a decrease in the cooling quality, bringing about a rapid temperature rise. Thus, the wear amount of CBFM with SCN increases significantly, especially for the friction disc in the axial middle position.

## 1. Introduction

The wet clutches are mainly used to transfer and cut off the torque from the engine to the axle, change gear, and locking-up components to prevent motion [1,2,3]. Typically, wet clutches used in heavy vehicles are under high energy density friction conditions which should possess stable and excellent friction-wear characteristics to ensure effective work performance and the long service life [4,5,6]. Thus, friction materials play an important role in the wet clutch which is required to have an adequate COF and a low wear rate [7,8,9]. The commonly used friction materials for friction discs are copper-based friction materials (CBFM) and paper-based friction materials (PBFM) [10]. Compared with PBFM, CBFM has better compact resistance, a steady COF, and excellent mechanical properties [11,12].

The CBFM is a kind of material that consists of the copper matrix, abradant, lubricant, and pores [13]. The copper matrix is the main component of CBFM, and the physical and mechanical properties of the matrix largely determine the overall friction-wear performance of friction materials [14]. The excellent physical and chemical properties of copper provide good thermal conductivity, stable friction performance, and low wear of CBFM [15].

A lot of work has been carried out in the literature aimed at improving the friction-wear properties of CBFM to make sure it has stable and excellent friction-wear properties with long service life. Ankur et al. [16] studied the copper-based composites fabricated by the powder metallurgy process and found the properties depend on some parameters such as reinforcement, particle size, the volume fraction of reinforcement, etc. Ho et al. [17] studied the effects of different fibers on the mechanical and tribological properties of copper semi-metallic friction material; it implied that the copper and brass fibers improved strengths significantly. Zhou et al. [18] constructed the friction and wear maps of copper matrix composites (Cu-MMCs), which demonstrated that the Cu-MMCs exhibited stable high COF and low wear rates when the iron volume content ranged from 10% to 15%. In addition, the result illustrates the dominant wear mechanisms as well [19]. Li et al. [20] presented a methodology to predict the wear status in the friction lining of wet clutch materials, indicating that the wear mechanism was related to the thermal degradation and mechanical effects. The abradant and lubricant had also received much attention from many researchers. Chen et al. [21] investigated the impacts of graphene on the microstructure, mechanical performance, and tribological properties of the composites. The result showed that graphene dramatically improved tribological performance. Celebi Efe et al. [22] studied the mechanical properties of copper which were enhanced by SiC particles. It indicated that the hardness of the composites increased with the amount and the particle size of particles. Scott et al. [23] studied the effect of the extreme pressure/anti-wear, viscosity index improver, and lubricating oil additives on the frictional characteristics of sintered copper alloy friction material. Guha Keshav et al. [24] added the steel fibers and fly ash into CBFM to develop a kind of friction material, in which the strength and hardness increased apparently. Jie et al. [25] studied the tribological properties changes of the carbon fabric wet clutch under the oil-lubricated condition, showing that the wear characteristic of carbon fabric composites exhibited a combination of adhesive wear, abrasive wear. Furthermore, some research focused on the effect of additives in lubricating oils as well [26]. Wu et al. [27] suggested that nanoparticles, especially CuO, added to oils could exhibit good friction-reduction and anti-wear properties.

Above all, the additives in CBFM have a vital impact on expanding the scope of application and developing the new friction materials on the wet clutch. Thus, the study of additives to improve the friction-wear properties of CBFM is necessary and urgent. The silicon carbide particles (SCN), which have the advantages of stable chemical properties, high thermal conductivity, low thermal expansion coefficient, and excellent wear resistance, have been widely used as an abrasive in composite materials [28,29,30,31]. It was employed to improve tribological properties such as anti-wear, reducing friction, and high load capacity [32]. Sap [33] investigated the microstructure, density, hardness, and wear behavior of Cu hybrid composites reinforced with TieB-SiCp powders, which indicated that the wear rate increased with the increasing reinforcement ratio. Some researchers focus on the tribology of nanoparticles themselves [34]. Jelita Rydel et al. [35] investigated the tribological properties of the modified Al_2_O_3_/SiO_2_ composite nanoparticles, found that anti-wear and anti-friction performances are significantly improved. The physical and chemical reaction between additives and matrix is also a necessary consideration. Jelita Rydel et al. [36] studied the ZDDP (Zinc dialkyl dithiophosphates) tribofilms on four different steel grades by a new AFM technique and analyzed the relationship between the steel microstructure and the tribofilm morphology. The results showed that the local ZDDP tribofilm thickness is affected by the microstructure of the underlying steel. Nevertheless, fewerresearch mentioned the effect of SCN on the friction-wear performance in CBFM. This study is aimed to make up for the deficiency.

Herein, the modified friction material (CBFM with SCN) formed by adding SCN in ordinary CBFM is tested in the SAE#2 bench. The effect of SCN on friction-wear properties is investigated experimentally in different operating conditions. The friction characteristics are reflected by the changes in torque and temperature. Moreover, the thickness variation, surface waviness, and roughness of friction pairs are analyzed to evaluate the wear characteristics of friction discs. The research results will provide new ideas for the selection and development of friction materials for wet clutches.

## 2. Materials and Methods

### 2.1. Test Samples

Although some analytical models, including coupling stiffness, material pairs, contact pressure, temperature, and shear rate were set up early [1], experimental data still constitute a necessary input for the function, especially the COF variables [37]. The SAE#2 clutch bench tests are conducted to simulate actual operating conditions of the wet clutch, in which the friction-wear data of the friction disc can be measured easily [38]. During the development of materials, the bench-top scan test to rank and select materials before execution in a full-scale test is economical and practical [39]. The structure and test standards of the SAE#2 bench have been described in previous studies [9,10,11].

As shown in Figure 1, the clutch pack is a six-friction-pairs system composed of four steel plates and three friction discs named P1, P2, and P3, respectively. The friction disc consists of a friction core and friction lining, and the steel plates and fiction cores are made of 65 Mn steel. The hardness of the steel plate ranges from 42 HRC to 48 HRC, the flatness is less than 0.20, and the roughness is required to be Ra0.8 to Ra1.25. Furthermore, it should be highlighted that there are two kinds of CBFMs. More exactly, the CBFM-A contains Cu(70 wt%), Sn(6 wt%), Zn(5 wt%), SiO_2_(5 wt%), Graphite(6 wt%), and some other additives (8 wt%). Based on A, a certain SCN is added to form the CBFM-B.

The spark plasma sintering processing technology is used to make the initial surfaces rough and porous to maintain a certain COF. The manufacturing process of the test sample is shown in Figure 2. As shown in Figure 1, The CBFM-A and CBFM-B have the same construction dimensions whose inner and outer diameters are 118.0 and 146.2 mm, respectively. The bisecting double-helical grooves, accounting for around 50% of the disc area, are designed in contact with the surface to guild oil and dissipate heat. The groove depth is about 0.3 mm, and 1.5 mm in width. The inner and outer diameters of the steel plate are 119.4 mm and 147.3 mm, respectively.

On account of the six-friction-pairs system, the temperatures of two thermometric plates are measured to regard as the clutch temperature, and there is a 6 mm hole in each plate to glue the K-type thermocouple.

### 2.2. Test Method

As listed in Table 1, the operating conditions are consisted of three operating parameters including the applied pressure, rotating speed, and automatic transmission fluid (ATF) temperature. As shown in Figure 3, during the SAE#2 test, the control and measurement system can collect four representative signals in one duty cycle, including the friction torque, applied pressure, disc temperature, and rotating speed, respectively. Based on the variation of friction torque, one engagement cycle consists of the filling phase, sliding phase, and sticking phase, respectively.

Before the test, the inlet lubricating oil with a flow velocity of 4 × 10^−3^ L/(min·m^2^) is heated up to the set temperature value. It needs to be emphasized that all the indexes of ATF, including the viscosity, flash point, and freezing point, are unified during the whole test. After that, the motor starts, and the rotating speed increase rapidly. When the rotating speed reaches the pre-set value, the motor powers off immediately; meanwhile, the pressure is applied on the clutch piston to push the friction discs to move axially against the back steel plate at *t_a_*.

Subsequently, the friction torque increases steeply and reaches *T_min_* at *t_b_*, the clutch temperature increases and the rotating speed decreases at the same time. The period from *t_a_* to *t_b_* is called the filling phase. The sliding phase is from *t_b_* to *t_d_*, including the stable working stage and rapid rising stage. The obvious feature of the stable working stage is that the friction torque remains in a relatively stable range, showing a flat curve in Figure 3. As the rotating speed decreases, the friction torque rises sharply from *t_c_* and reaches the maximum value (*T_max_*) at *t_d_*, which is the rapid rising stage. The period after *t_d_* is the sticking phase. Each engagement cycle is continuously repeated five times, and the measurement system stores one of the experimental data as the result randomly.

### 2.3. Characterization of the Friction-Wear Properties

Supposing that the applied pressure is uniformly distributed on the friction surface, the instantaneous COF *μ(τ)* can be figured by [40]:(1)$$\mu (\tau )=\frac{T(\tau )(r_{O}^{2}-r_{i}^{2})}{2zF(\tau )(r_{o}^{3}-r_{i}^{3})}$$
where *z* is the number of friction pairs; *T* is the friction torque; *F* is the axial force on the friction surface (N); *r*_i_ and *r*_o_ are the inner and outer radii (mm), respectively.

Furthermore, the mean value of the instantaneous COF is primarily important to evaluate the global frictional performance. In general, when the mean COF is smaller than a certain value, the engagement time will be prolonged, even worse the engagement will be a failure. The mean COF can be calculated as follows [40].
(2)$$\mu_{a}=\frac{1}{t_{c}-t_{b}}\int_{t_{b}}^{t_{c}}\mu (\tau )\;d\tau$$

However, the mean COF fails to describe the changing trend accurately. Thus, the variable coefficient is used to describe the slope of the friction torque in the stable working stage as follows,
(3)$$\alpha =\frac{T(t_{c})-T(t_{b})}{t_{c}-t_{b}}\times 100\%=\frac{\mu \left(t_{c}\right)-\mu \left(t_{b}\right)}{t_{c}-t_{b}}\times 100\%$$
where the *T*(*t_c_*) and *T*(*t_d_*) are the friction torques at *t_c_* and *t_d_*, respectively. *μ*(*t_c_*) and *μ*(*t_d_*) are instantaneous COF at *t_c_* and *t_d_*.

Additionally, the average thicknesses of the friction discs before and after the test are measured by a micrometer. The surface morphology of friction discs is obtained by the GTX-20-1014 (the white light interferometer made in Germany) White-light interferometer. The areal height deviation of the surface is given by the morphology parameters *S_a_*, *S_q_*, the maximum pit height of the surface (*S_v_*), the maximum height of the surface (*S_z_*), and the maximum surface peak height (*S_p_*).

The *S_a_* is the arithmetic mean of the absolute value of the height within a sampling area, thus,
(4)$$S_{a}=\frac{1}{A}\int \left|z(x,y)\right|dxdy.$$

The *S_q_* is defined as the root mean square value of the surface departures, thus,
(5)$$S_{q}=\sqrt{\frac{1}{A}\iint_{A}z^{2}(x,y)dxdy}$$
where *A* is the sampling area.

The rectangular area with a length of 1.2 mm and a width of 0.9 mm on the friction plates CBFM-A and CBFM-B is selected to obtain the pre-test and post-test morphologies on the white light interferometer with a magnification of 50 times.

## 3. Results and Discussion

### 3.1. Friction Characteristics

#### 3.1.1. Applied Pressure

The friction torque curves of CBFM-A and CBFM-B under the different applied pressures ranging from 0.3 MPa to 2 MPa are shown in Figure 4. At the early slipping phase, the friction torques of CBFM-A and CBFM-B both fluctuate severely. This is mainly due to the changes in contact status. The oil films can be easily formed on the contact surface at the filling phase; as the applied pressure is loaded, the films are squeezed seriously. When the applied pressure becomes stable, the asperities contact dominates the engagement process, which not only hinders the formation of oil films but also accelerates theirs cracking. Thus, changes of contact status between friction pairs lead to violent fluctuations in friction torque curves.

After that, the friction torque keeps in a relatively stable stage. In this stage, the friction torque of CBFM-B is higher than that of CBFM-A under all preset pressures. More exactly, as shown in Figure 4a, the mean friction torque of CBFM-A is 19 N m, and this of CBFM-B is 31 N m. Otherwise, *T_max_* and *T_min_* of A are 66 and 34 N m, and these of CBFM-B are 88 and 53 N m. In general, the friction torques of CBFM-A and CBFM-B both increase apparently with the applied pressure. Furthermore, the variation of friction torque with engagement time is diverse in this stage. As shown in Figure 5a, the variable coefficients of A are −3.46, −3.24 under 0.3 and 0.6 MPa, respectively, and the values of CBFM-B are −6.34 and −2.67. The variable coefficients are below zero which indicates that the friction torques tend to decrease in the stable working stage. It can be noted that the variable coefficients of CBFM-B rise faster than that of CBFM-A, as shown in Figure 5a. Overall, the friction torque of CBFM-B is significantly higher than that of CBFM-A in the experiment. The variable coefficients of CBFM-A and CBFM-B both increase with applied pressure steeply, but the increase of CBFM-B is sharper. The features of friction torque and the variable coefficient can be given the reasons as follows.

The friction torque is dependent on the summation of the contact torque and hydrodynamic torque. Since the thickness of oil films is relatively greater under the low applied pressure, the hydrodynamic torque dominates the contact process. With the engagement processing, the clutch temperature increases inevitably, decreasing the oil viscosity, resulting in the decrease of the hydrodynamic torque naturally. With the increase of pressure, the contact torque dominates categorically which makes the changes of the hydrodynamic torque appear slightly; thus, the variable coefficients under the high-pressure increase slightly.

The curves of the temperature corresponding to the different applied pressures are also shown in Figure 4. Except for the 0.3 MPa condition, the temperature increases gradually and reaches the maximum value at *t*_d_. As shown in Figure 4a, the temperatures of CBFM-A show a decreasing trend after an increase. The reason for the change can be explained as follows. The clutch temperature is dependent on the generated friction heat and the heat dissipation of ATF. As the engagement processes, the disc temperature increases naturally. However, with the rotating speed decreasing rapidly, the generated heat is reduced progressively; simultaneously, the heat dissipation capacity changes a little. It results in a decrease in disc temperature. Furthermore, the increased rates and the maximum temperatures of CBFM-B are significantly higher. Furthermore, the generated temperature is related to the applied pressure and the mean COF. As shown in Figure 5b, the mean COF of CBFM-B is 30% to 50% higher than that of CBFM-A. The higher mean COF of B results in a greater temperature rising rate eventually. Thus, the temperature rising rate of B, the slope of the temperature curve, increases from 65 at 0.3 MPa to 298 at 2 MPa. At the same time, the higher mean COF brings about a shorter engagement time which is only 0.41 s at 2 MPa condition. As a result, the SCN improves the COF of friction materials indeed.

#### 3.1.2. ATF Temperature

ATF temperature is a vital factor affecting the clutch friction performance. As shown in Figure 6, the friction torques of CBFM-B are also higher than that of CBFM-A under temperature conditions. More exactly, as the ATF temperature increases, the mean friction torques of CBFM-A are 108, 96, 88, and 85 N·m. The mean friction torques of CBFM-B are 145, 138, 136, 135 N·m under the same conditions. The difference is also reflected in the maximum values of friction torque. The *T**_max_*** of CBFM-A are 460, 460, 420, and 377 N·m, respectively, and these of B are 430, 451, 395, and 374 N·m. It also can be noticed that the *T_max_**s* of CBFM-A and CBFM-B decrease gradually with the ATF temperature. As shown in Figure 7a, the variable coefficients of CBFM-A are from −18.62 to 31.62, and this of CBFM-B increases from −24.35 to 52.25. It means that the variable coefficients of CBFM-A and CBFM-B increase expeditiously with the rise of ATF temperature. Although the value of CBFM-A is high than CBFM-B at a temperature below 80 °C however, the variable coefficient of CBFM-B is significantly higher than that of CBFM-A as the ATF temperature increases. Thus, the results can be concluded that the friction torque of CBFM-A and CBFM-B are both affected by ATF temperature, and the change rate of CBFM-B is faster indeed.

As shown in Figure 6, the temperature increase rates of CBFM-A are 16.53, 14.87, 13.10, and 11.42, respectively. In addition, these of CBFM-B are 19.46, 20.12, 18.63, and 19.71. It can be noticed that CBFM-B has higher temperature increase rates than CBFM-A. Thus, the changes in ATF temperature will result in the violent temperature increase of CBFM-B, which leads to acute wear at high temperatures. It demonstrates that the thermal stability of CBFM-B is worse than CBFM-A.

As shown in Figure 7b, under different ATF temperature conditions, the mean COF of CBFM-B is about 36% to 45% higher than that of A. With the increase of ATF temperature, the mean COFs of CCBFM-A and CBFM-B decrease slightly and both reach the minimum value at 100 °C. It can be interpreted as follows. As the ATF temperature increases, the viscosity and thickness of ATF decrease quickly. Thus, the dominating hydrodynamic torque decreases, which causes the mean COF to decrease firstly. As the ATF temperature increases continually, the asperities of the friction surface are more likely to break through the ATF film to generate the asperity contact, then bringing about the worse lubrication status. To be more exact, the lubrication status transfers from the mixed lubrication to the boundary lubrication. Thus, the mean COF increases eventually.

#### 3.1.3. Rotating Speed

Figure 8 demonstrates the friction torques and disc temperatures of CBFM-A and CBFM-B at different rotating speeds. The friction torques in the early slipping phase of CBFM-B show an upward trend, even without a stable working stage, for rotating speeds below 1500 rpm. However, the friction torque of CBFM-A is stable and long-lasting with little fluctuation. Thus, the friction stability of CBFM-B is worse under a lower rotating speed. As shown in Figure 9a, the variable coefficient of CBFM-A and CBFM-B decreases obviously. The value of CBFM-A decreases from 282 to 32, and that of CBFM-B decreases from 674 to 52. As the rotating speed increases from 600 to 1500 rpm, the change is most obvious, which of CBFM-A is from 284.42 to 45.62. At the same time, the variable coefficient of CBFM-B changes from 676.51 to 177.58. This confirms that the friction stability of A is much better at different rotating speeds. The specific reasons are as follows. Under the low rotating speed condition, the ATF film is thin enough that the friction torque is dominated by the contact torque. Thus, the friction torques of CBFM-A and CBFM-B both increase quickly. Since the porous surface of CBFM-A can store more oil to form a thicker oil film during the engagement process, the friction stability of CBFM-A is far better. With the rotating speed increasing, the hydrodynamic torque dominates gradually at the early filling phase, thus the stable working stage appears at a high rotating speed. However, as the engagement processes, the asperity contact dominates the friction again.

The mean COFs of CBFM-A and CBFM-B both decrease first and then increase; when the rotating speed is above 1500 rpm. It should be emphasized that even if there is the same change trend, the COF of CBFM-B is higher than that of CBFM-A. It can also be explained by the change of hydrodynamic friction. The higher the mean COF is, the more the generated friction heat is. Consequently, the temperature increase rate of CBFM-B is much higher. Moreover, the engagement times of CBFM-A increase from 0.56 to 1.61 s; and that of CBFM-B are from 0.48 to 1.20 s. Due to the worse friction stability and less engagement time at low speed, the CBFM-B is more suitable for use at high speed, in which the clutch needs much better cooling conditions.

### 3.2. Wear Characteristics

#### 3.2.1. Thickness Variations

Wear always accompanies the process of friction. Thickness variation is an important index to evaluate the wear degree. The thicknesses of the pre-test discs and post-test discs are shown in Figure 10. There are 35 measuring points uniformly distributed in the circumferential direction. As shown in Figure 10a, the curves are tortuous and fluctuating, suggesting that the contact surface of CBFM-A is spiky. Nevertheless, as shown in Figure 10b, the thickness curves of CBFM-B are straighter and smoother, thus the contact surface of CBFM-B is homogeneous. The micro-contact area of CBFM-B accounts for a larger proportion which results in the higher friction torque of CBFM-B in the engagement process eventually.

The wear degree of friction discs in different axial positions is quite different as well. The pre-test and post-test thickness measurements of the samples are shown in Table 2 and Figure 10c. The mean thickness of the P2 disc is significantly smaller than the others. As shown in Figure 10d, the wear amount of P2 shows the same pattern. It was observed that the wear degree of CBFM-B is higher, and the P2 discs of both CBFM-A and CBFM-B are worn more seriously than the others.

#### 3.2.2. Micro-Morphology Variations

The white light interferometric profilometry and surface waviness of pre-test discs in CBFM-A and CBFM-B are shown in Figure 11. The blue and green areas are pores with a depth around 40 μm, and the black area is the one with a depth over 40 μm. Both the depth and size of the pores in CBFM-A are significantly higher than those in CBFM-B. Furthermore, the blue and green areas of CBFM-B occupy a smaller proportion of the scanning area. This intuitively reflects the difference between the surface topographies and indicates that the compactness of CBFM-B is better, which has fewer and smaller surface pores.

Otherwise, the morphology parameters *S_a_*, *S_q_ S_v_*, *S_z_*_,_ and *S_p_* of pre-test surfaces of CBFM-A and CBFM-B are shown in Table 3. The *S_a_* and *S_q_* are considered to reflect the surface roughness well. The *S_p_*, *S_v_*, and *S_z_* parameters give absolute values for features on the surface which can be used in conjunction with the *S_a_* and *S_q_* to describe surface topography more comprehensively. Comparing the values of the samples, it is obvious that the surface roughness of CBFM-A is higher, which is related to the addition of SCN. At the same time, the variations of surface waviness paralleled the *x* and *y* axes expound the same rule as well.

The post-test white light interferometric profilometry of CBFM-A and CBFM-B are shown in Figure 12 and Figure 13. Compared with the pre-test surfaces in Figure 11, the post-test surfaces are more flattened. The reason for this is the peaks are cut during the test process. Thus, the changes in roughness can be calculated by the wear degree as well as thickness.

The morphology parameters of the post-test surfaces, reflecting the roughness of materials, are shown in Table 4. For example, the *S_a_* of disc P1, P2, and P3 with CBFM-A are 1.054, 0.870, and 1.325 μm, respectively, and these of CBFM-B are 0.657, 0.379, and 0.547 μm, respectively. It shows that the surface roughness of CBFM-B is significantly lower than CBFM-A. The surface waviness expresses the roughness fluctuations on the axis in the scanned area, shows the same rule. The results express that the post-test surface is not consistent either. Compared with CBFM-A, there are fewer extremely high peaks and low valleys in the contact surface of CBFM-B, which indicates that the wear degree of CBFM-B is serious.

Furthermore, Comparing the morphology parameters of different friction discs in CBFM-A or CBFM-B, it can be seen that the values of P2 are significantly lower than those of P1 and P3. Otherwise, the metallographic diagrams of the steel plates corresponding to the P2 discs are shown in Figure 14. The steel plates have obvious scratches, as shown in Figure 13a. However, the steel plates of CBFM-B appear the phenomenon of copper-transfer, as shown in Figure 13c, a large amount of friction material is bonded to the steel plates. It means that material transfer occurs on the friction disc during the test process. Moreover, the length, depth, and several scratches in CBFM-B are significantly higher than those in CBFM-A. Above all, the conclusion can be made that the wear degrees of different discs are not consistent, and the wear degree of P2 is more serious.

Compared to the post-test graphs of CBFM-A and CBFM-B, the pores in CBFM-A are far bigger than these of CBFM-B as well. The pores are mainly composed of two parts, one is the pores on the initial surface, and the other is the furrow and spalling dug by the wear debris during the test. It further shows that the porosity of CBFM-A is higher than that of CBFM-B, and the depth and area of the pores are much larger than that of CBFM-B. The less porous contact surface leads to the thin oil film of CBFM-B, which makes the contact torque dominates early. The dominance of the contact torque results in a higher friction torque for CBFM-B at all pre-set operating conditions. The porous and spiky surface of CBFM-A results in more ATF in pores, thus its heat dissipation performance is much better. The morphological differences can be explained as follows.

As shown in Figure 15, during the sintering and molding process of the material, the SCN refines the grains, resulting in a decrease in the surface roughness, as shown in Figure 15b. At the same time, the SCN also causes stress concentration between grains, which leads to is more serious wear under the action of friction. Firstly, due to the high roughness of the CNFM-A surface, more lubricating oil can be stored. Secondly, the grain size is relatively large, and there are few cracks between grains. The wear debris’ number of CBFM-A is small, but the size is relatively large, as shown in Figure 15c. However, due to the addition of SCN, the wear debris’ size and number of CBFM-B are just opposite, as shown in Figure 15d. Combining the changes in thickness, surface roughness, and porosity before and after the test, it can be concluded that the wear of CBFM-B is more serious. Consequently, the addition of SCN reduces the surface porosity of friction material, leading to increased wear degree ultimately.

## 4. Conclusions

The results from this study are as follows.
The CBFM with SCN exhibits a higher friction torque, compared with CBFM, the COF is increased by 30% to 50%. It suggests that the SCN additives can help to increase the COF of CBFM. At the same time, as the temperature rising rate of CBFM with SCN is quite greater, rising from 65 at 0.3 MPa to 298 at 2 MPa, the thermal stability of CBFM is reduced by adding SCN indeed.The variable coefficient increases significantly with the rise of pressure and temperature, while the change in the variable coefficient of CBFM with SCN is higher than that of CBFM. Meanwhile, the variable coefficient of CBFM with SCN decreases from 674 to 52 with the growth of rotating speed; the magnitude of change is much greater than CBFM. Thus, the friction stability of CBFM with SCN is worse.Since the SCN changes the compactness and microscopic morphology of CBFM, the initial microscopic morphology of the CBFM with SCN is flatted with a less porous area. Thus, the SCN additive brings about more serious wear of friction components, especially for the P2 disc.The research results can provide the theoretical basis for the development of new friction materials with enhanced friction-wear properties.

## Figures and Tables

**Figure 1 materials-15-00587-f001:**
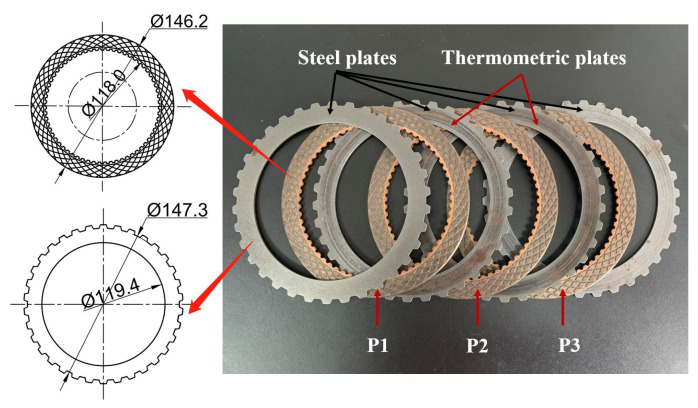
Dimensions of friction discs and steel plates.

**Figure 2 materials-15-00587-f002:**
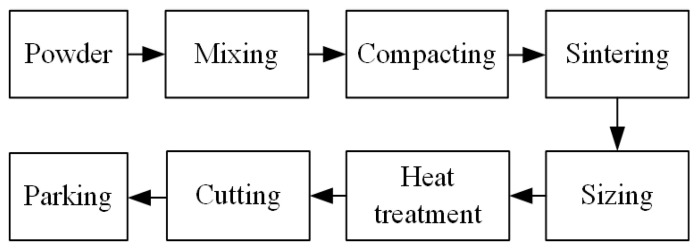
The manufacturing process of the test sample.

**Figure 3 materials-15-00587-f003:**
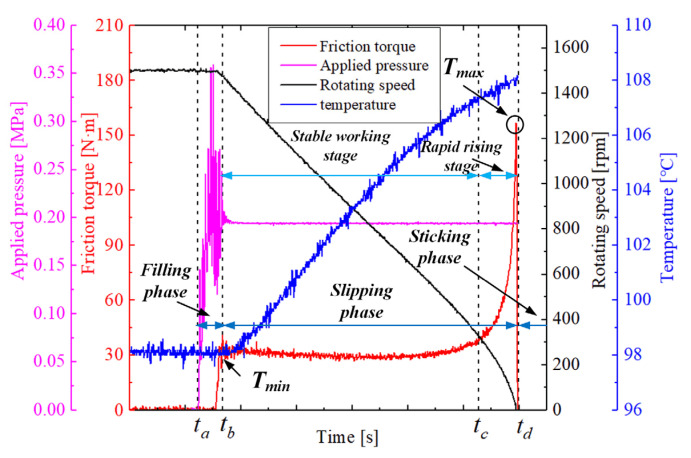
Representative measurement signals.

**Figure 4 materials-15-00587-f004:**
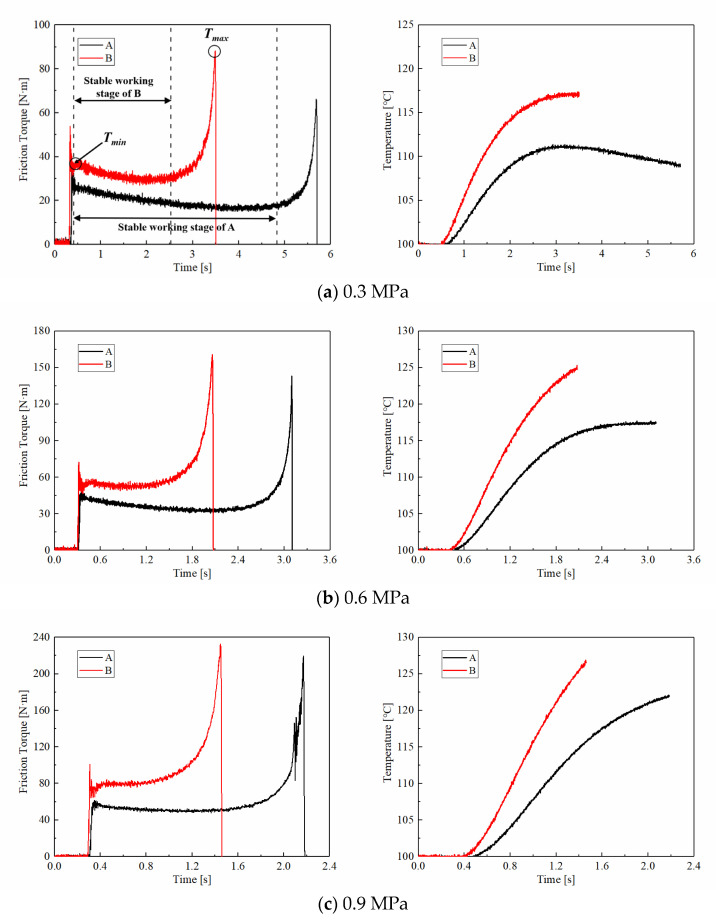
Variations of the friction torques and surface temperatures of CBFM-A and CBFM-B under different applied pressures (**a**–**f**).

**Figure 5 materials-15-00587-f005:**
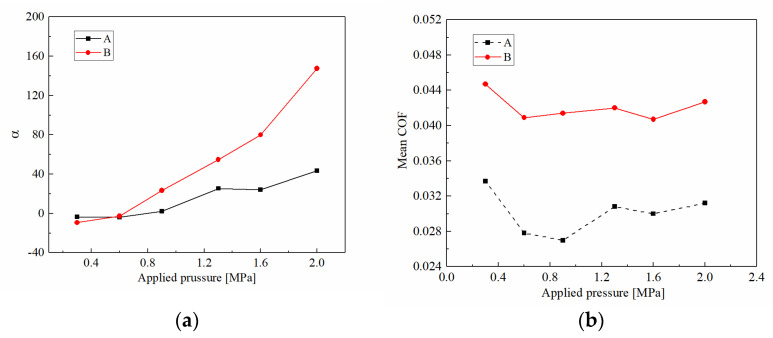
Variable coefficients in stable working stage (**a**) and mean COFs (**b**) of CBFM-A and CBFM-B under different applied pressures.

**Figure 6 materials-15-00587-f006:**
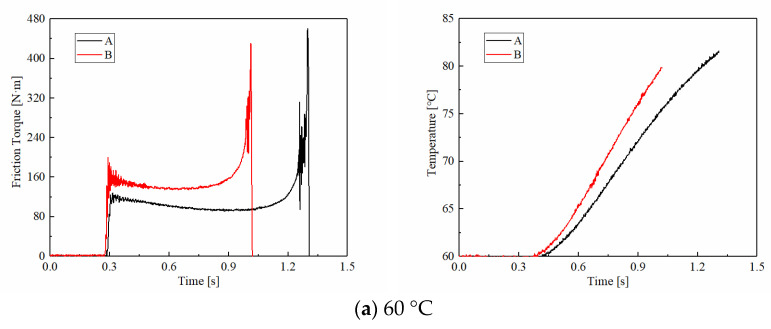
Variations of the friction torque and surface temperatures of CBFM-A and CBFM-B. (**a**) 60 °C, (**b**) 80 °C, (**c**) 100 °C, (**d**) 120 °C.

**Figure 7 materials-15-00587-f007:**
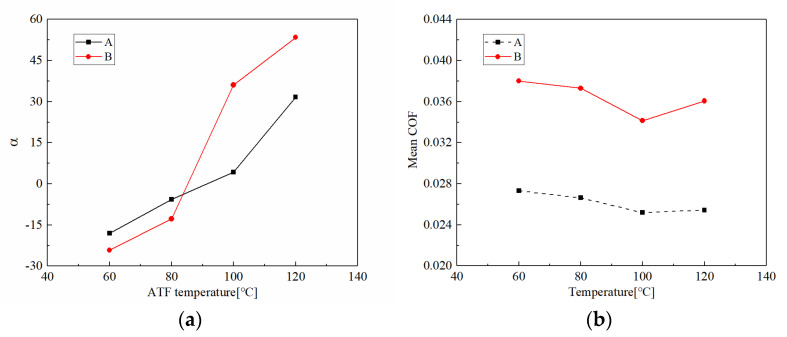
The variable coefficients in stable working stage (**a**), and friction torques of CBFM-A and CBFM-B in different ATF temperatures (**b**).

**Figure 8 materials-15-00587-f008:**
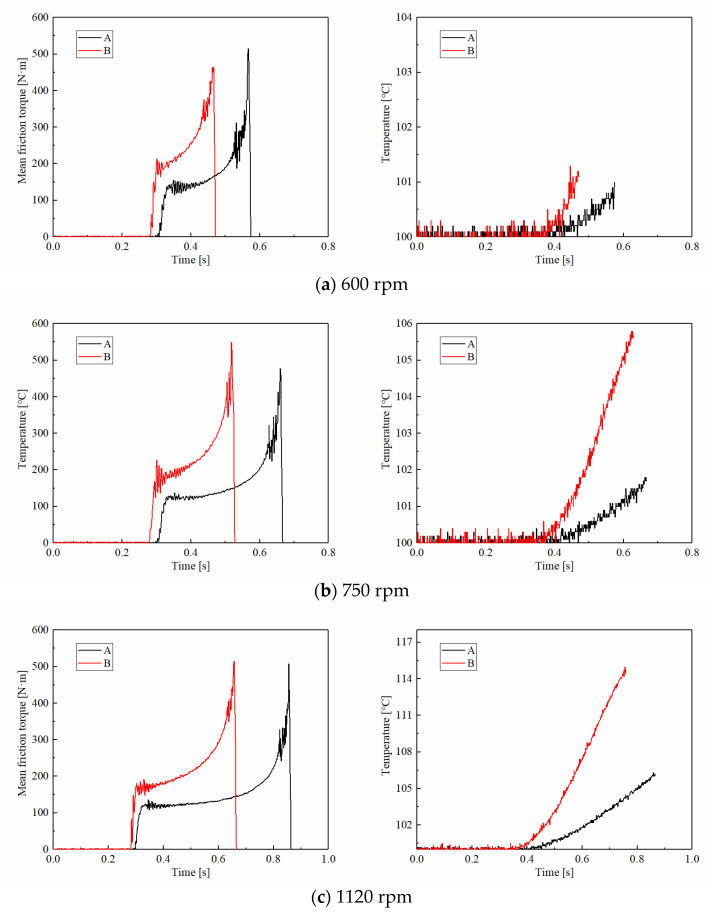
Variations of the friction torque with rotating speeds of CBFM-A and CBFM-B (**a**–**f**).

**Figure 9 materials-15-00587-f009:**
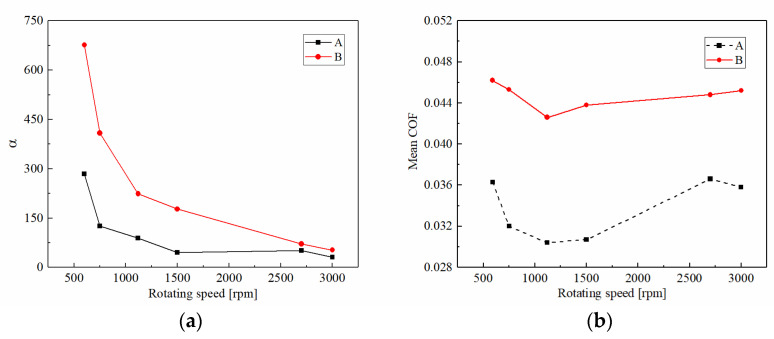
The variable coefficients of friction torque in stable working stage (**a**) and the mean COF of CBFM-A and CBFM-B (**b**) at different rotating speeds.

**Figure 10 materials-15-00587-f010:**
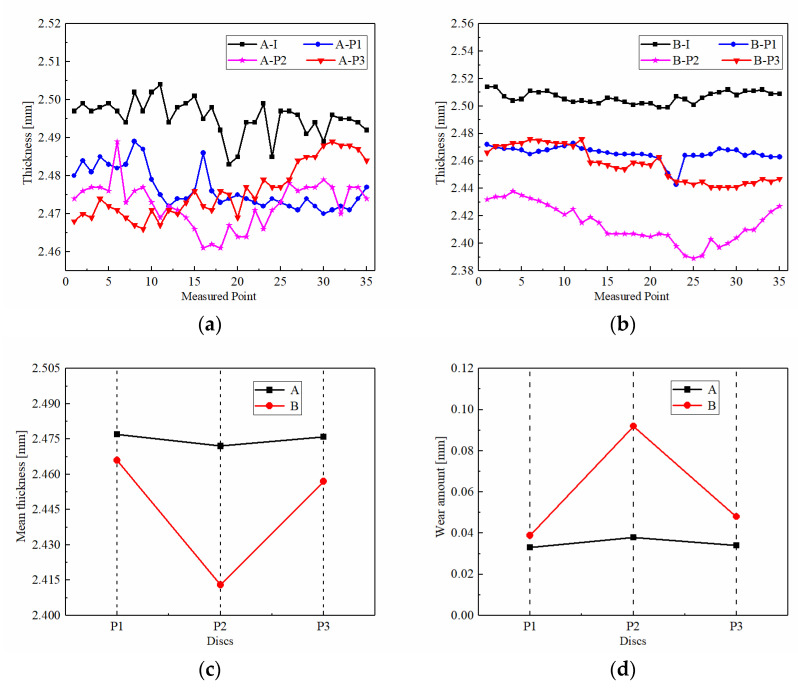
A-I is the pre-test measured thickness of CBFM-A; A-P1, A-P2, and A-P3 are the post-test measured thicknesses (**a**). B-I is the pre-test measured thickness of CBFM-B; B-P1, B-P2, and B-P3 are the post-test measured thicknesses (**b**). The mean thicknesses of discs (**c**) and the wear amounts (**d**).

**Figure 11 materials-15-00587-f011:**
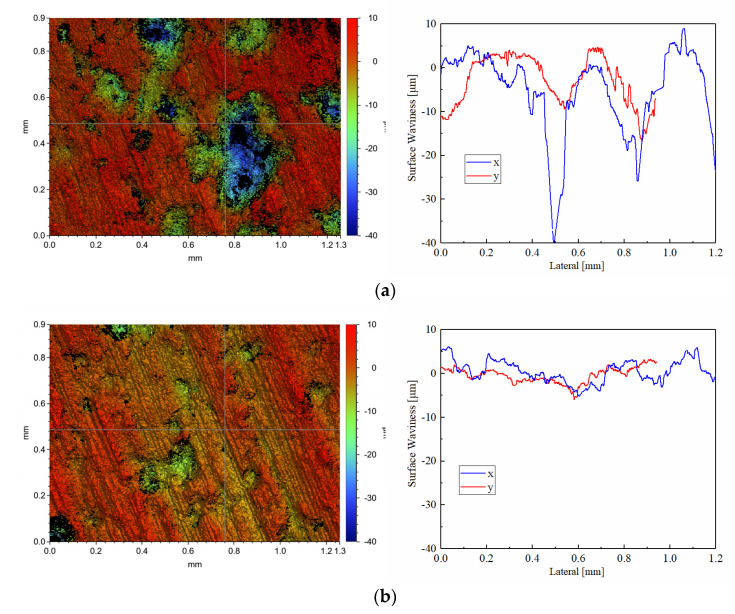
The white light interferometric profilometry (×50) and surface waviness of pre-test in CBFM-A (**a**) and CBFM-B (**b**).

**Figure 12 materials-15-00587-f012:**
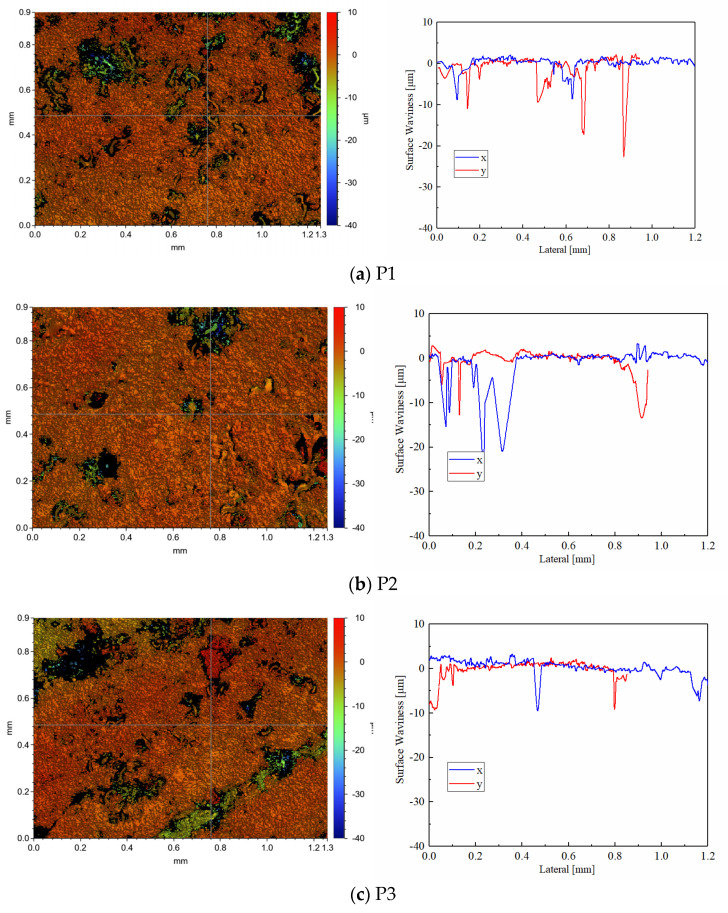
The post-test white light interferometric profilometry (**a**–**c**) (×50) of the test sample CBFM-A.

**Figure 13 materials-15-00587-f013:**
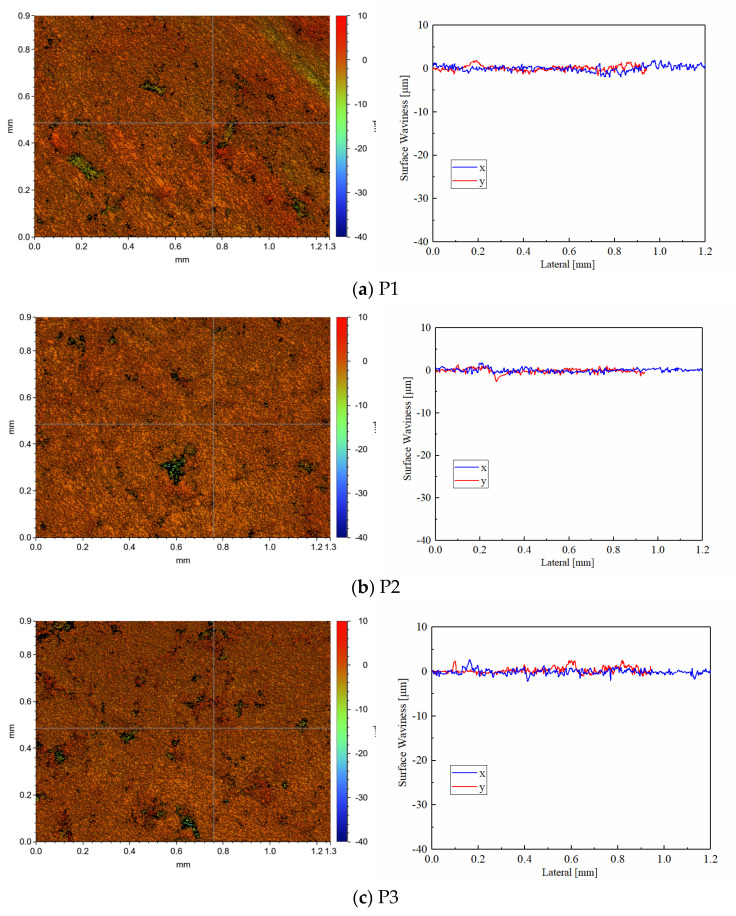
The post-test white light interferometric profilometry (**a**–**c**) (×50) of the test sample CBFM-B.

**Figure 14 materials-15-00587-f014:**
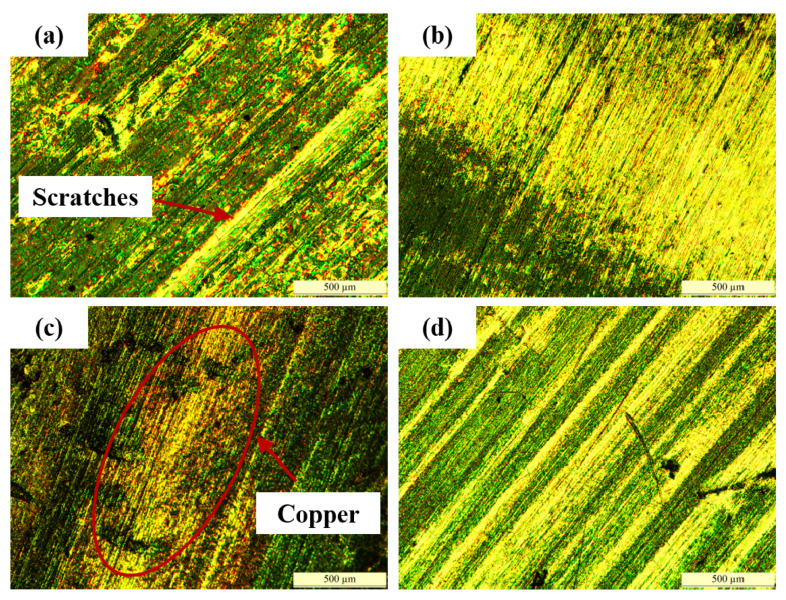
The metallographic diagrams (×50) of the steel plates correspond to the P2 discs of CBFM-A (**a**,**b**) and CBFM-B (**c**,**d**).

**Figure 15 materials-15-00587-f015:**
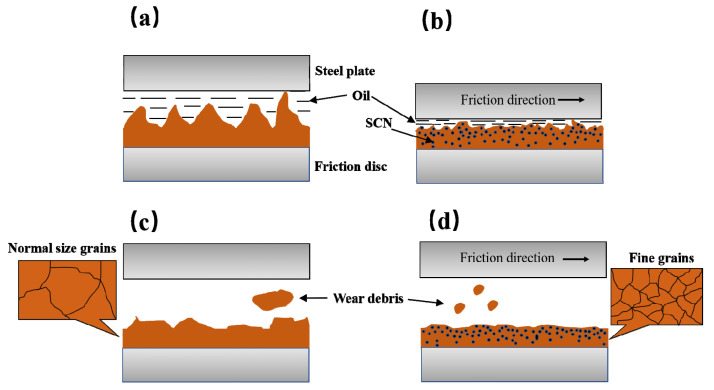
Schematic diagram of friction surface morphology, the pre-test surface of CBFM-A (**a**) and CBFM-B (**b**), the post-test surface of CBFM-A (**c**) and CBFM-B (**d**).

**Table 1 materials-15-00587-t001:** Operating conditions.

Parts	Applied Pressure (N·m)	Rotating Speed (rpm)	ATF Temperature (°C)
1	0.3, 0.6, 0.9, 1.3, 1.6, 2.0	1500	100
2	1.6	1500	60, 80, 100, 120
3	2.0	600, 750, 1120, 1500, 2700, 3000	100

**Table 2 materials-15-00587-t002:** Mean thicknesses of discs in CBFM-A and CBFM-B.

	CBFM-A		CBFM-B	
	Pre-Test (mm)	Post-Test (mm)	Pre-Test (mm)	Post-Test (mm)
P1	2.510	2.477	2.505	2.466
P2		2.472		2.413
P3		2.476		2.457

**Table 3 materials-15-00587-t003:** Morphology parameters of pre-test surface of CBFM-A and CBFM-B.

Amplitude Parameters	CBFM-A	CBFM-B
*S_a_* (μm)	5.293	2.278
*S_q_* (μm)	7.547	2.885
*S_p_* (μm)	12.832	9.008
*S_v_* (μm)	−56.321	−23.662
*S_z_* (μm)	69.153	32.669

**Table 4 materials-15-00587-t004:** Morphology parameters of the post-test surface of CBFM-A and CBFM-B.

	CBFM-A			CBFM-B		
Amplitude Parameters	P1	P2	P3	P1	P2	P3
*S_a_* (μm)	1.054	0.870	1.325	0.657	0.379	0.547
*S_p_* (μm)	14.061	12.200	38.814	8.876	5.408	8.861
*S_q_* (μm)	2.295	2.003	2.350	0.982	0.641	0.899
*S_v_* (μm)	−40.060	−38.287	−46.319	−32.559	−21.911	−25.171
*S_z_* (μm)	54.121	50.487	89.845	41.435	27.319	34.032

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
