# Peer review of "Effect of Silicon Carbide Nanoparticles on the Friction-Wear Properties of Copper-Based Friction Discs"

_materials, 2022, doi:10.3390/ma15020587_

Round 1
Reviewer 1 Report
In this study, the researchers investigated the effect of nano-sized SiC particles on the friction-wear properties of copper-based clutch material. The experimental setup according to the SAE#2 test was well organized. However, some points that I think weaken the scientificness of the study are listed below.
- Important and remarkable numerical data can be added to the abstract and conclusions sections of the study.
- The researchers propose adding SiC particles into the copper-based clutch matrix. However, the first question that comes to mind in this regard is whether the hard SiC particles released by the wear of the clutch will damage the gearbox elements. Because the SiC particle, which is a hard ceramic, has abrasive characteristics. Also, do these SiC particles affect the properties of the lubricating oil? Researchers should briefly discuss these issues.
- In the study, no information was given about how copper based friction materials were produced and what their SiC ratios were. I think it is important to provide these information. In addition, agglomeration is a major problem during the adding of nano-sized particles into the matrix. In this regard, I strongly recommend that researchers use the following reference in the study. How do researchers avoid this agglomeration problem? Is agglomeration a problem with this material?
Gok M.G., “Spark Plasma Sinterıng of Nano Silicon Carbide Reinforced Alumina Ceramic Composites”, European Mechanical Science, 5, 2, 64-70, (2021). Doi: https://doi.org/10.26701/ems.841961
- In the results and discussion section, researchers should verify or criticize their results by comparing them with the literature data. In this way, I think that the scientific aspect of the study will increase even more.
Reviewer 2 Report
REVIEW
ABSTRACT:
- The abstract should begin with a clear statement about the purpose or purpose of the study.
- It should not only state what was done, but also why it was done.
- The summary should contain results that quantitatively reflect the improvement in wear behavior of the nanoparticle addition compared to copper-based friction materials (CBFM).
- The summary should conclude with a summary of the important contributions of this research to understanding the nature of wear in materials.
- The summary section needs to be rewritten.
- INTRODUCTION
- The sentence starting with “Many studies so far” should be removed from the article. Because, “Typically, wet clutches used in heavy vehicles are under high-energy-intensity friction conditions that must have stable and excellent friction-wear characteristics to provide effective operating performance and long service life [4-6]” reveals the purpose of the studies in the literature.
- This sentence states, “A lot of work has been done in the literature aimed at improving the friction-wear properties of CBFM (to ensure that it has stable and excellent friction-wear properties to provide effective operating performance and long service life).” it should be in shape. In this way, fluidity is maintained.
- Again, with reference to the studies in the literature, it is emphasized that the hybrid copper-based hybrid composites formed by adding ceramic particles provide significant positive benefits in the mechanical and wear characteristics, and the necessity of the study is revealed and the wear resistance success of the SCN addition is proven with the results of the cited studies. The authors are expected to revise the introduction of the article by referring to the improvements in the wear behavior and mechanical properties of the ceramic-doped Metal Matrix Composites in the studies whose doi is given.
- “However, less research has mentioned the effect of nanoparticles in CBFM.” general expression is used. This statement should emphasize “what effect” has on the friction-wear properties in CBFM. For this purpose, the ceramic particles to which SCN nanoparticles belong should be supported by other studies in powder metallurgy in the literature. Study doi should be supported by reference to the results of these recent studies.
- “Research results can provide a theoretical basis for the development of new friction materials with improved friction-wear properties.” The sentence should be moved in the conclusion.
- Materials and Methods
2.1. test samples
- It is appropriate to use sentences emphasizing that “Although there are analytical models that include coupling stiffness, material pairs, rotor/stator contact pressure, temperature, shear rate and COF, experimental data still constitute a necessary input for COF, especially as a function of other variables” will be.
- (Any OEM-specific test protocol run on an SAE No. 2 Friction Testing Machine is typically used to obtain such data. During material development, with a simple bench-top scan test to rank materials prior to execution in a full-scale test saves both cost and time.SAE No. 2 includes the clutch material itself, the reaction plate, and the fluid with which the clutch works, in this case called automatic transmission fluid (ATF).) similar content can be shared to educate the reader on the subject.
- No nanostructure analysis or information about particle size and morphology has been shared for SCN powder particles. The size and morphology of the particles affect the mechanical properties and wear behavior of the test specimens.
- Has any spectral analysis been performed for the powder mixture prepared with SCN addition or for the abrasion samples produced?
- If an analysis was made with either elemental mapping of the powder mixture in the composition and any of the EDX or XRD methods proving its presence, information in this section
- Information should be given whether volumetric or mass ratio was used in the elemental distribution in the composition of the test samples with SCN addition.
- As stated in the introduction, the mechanical properties of the wear elements are important on the tribological behavior of the wear elements. Has any test or analysis been carried out to determine the mechanical properties of the SCN added test samples, which are the subject of this study?
- Considering the information in the literature above; No information was shared about the mechanical properties of the CBFM test samples. If CBFM test specimens are commercially available, information on structural properties such as density and porosity or important mechanical properties such as hardness and tensile strength should be shared on the firm's data sheet.
- No information was shared about the mechanical properties of the SCN test samples. Have any tests been carried out on the structural properties of the SCN test specimens such as density and porosity, or on important mechanical properties such as hardness, tensile strength? If it has been done, information about the methods used should be shared in this section.
- No information has been shared for the properties of ATF oil used in wear tests. First of all, information reflecting the typical characteristics of ATF should be shared.
- Again, the numerical value of the viscosity index of ATF oil causes significant effects on the wear behavior of tribo pairs in oil film thickness and wear tests. For this reason, this effect was never mentioned when interpreting the wear test results.
- Wear does not occur by itself, but by tribo pairs. For this reason, micrographs of the reaction discs (from 65Mn steel), CBFM and SCN added wear samples should be taken in SEM or optical microscope in order to perform the wear status, occurring wear mechanisms, dominant wear mechanism and surface morphological analysis. If received, it must be shared. This helps to support interpretations and increase the validity of the study.
2.3. Characterization of the friction-wear properties
- If formulas (1), (2) and (3) are used to define instantaneous COF μ(τ), Average COF average μ(a), variable coefficient to define the slope of the changing trend of COF, this should be expressed clearly.
- If these values are read through the interface on the PC in the test setup, unnecessary literature is theoretical knowledge and these formulas should be deleted. Sharing theoretical knowledge in the unnecessary literature does not contribute to the understanding of the study.• Increasing interface temperature lowers the oil viscosity and subsequently causes a reduction in shear stress produced by the oil film at high junction pressure.
- “The same proprietary processing technology is used to make initial surfaces rough and porous to maintain a given COF.” Based on the expression, how many stages do these special processing technologies consist of and which processing methods do they cover? Are there any standards regarding this? Creating mysterious situations for the reader is undesirable in scientific studies. Authors are required to share detailed information about this processing method.
- No information about the size of the sampling area, A taken with the interferometer, was shared.• White light interferometric profilometry (Figure 3), showing the first surface morphologies of the test sample with CBFM and SCN addition, is a result and should be moved to the “Results and discussion” section. Necessary evaluations should be included in this section.
- The values shared here for the morphology parameters Sa, Sq, Sv, Sz and Sp used in the evaluation of the first surface topography index of the CBFM and SCN added test samples are a result and these values should be conveyed in the Results and discussion section.
- It is seen from the graphs that the points where the engagement is unstable form hysterical curves. At these points, these instabilities cause vibration in the test setup. At this point, this vibration value creates a negative effect for the oil film. Excessive vibration causes the oil film to rupture, which changes the type of interaction between tribo pairs. For this reason, has any vibration analysis been carried out? The effect of this situation is not included in the conclusion part.
- “It has been found that μd decreases with increasing clutch pressure due to reduced mechanical locking of asperities and oil film shear stress.”
- Results and discussion
3.1. Friction characteristics
3.1.1. Applied pressure
- “In the early slip phase, the friction torques of both A and B fluctuate considerably. This is mainly due to changes in communication status.” Based on its determination, this situation will be reflected as vibration for the gearbox [doi.org/10.1016/j.triboint.2013.08.003]. At this point, this vibration is not mentioned at all. Comfort and other structural requirements in cars make vibration control and optimization difficult.
- “B's friction torque is higher than that of A under all preset pressures. More precisely, as shown in Figure 4(a), the Tmax and Tmin of A are 66N·m and 34N·m, and for B these are 88N·m and 53N·m.” Its detection is a reflection of an improvement in terms of mechanical properties. Throughout the entire text, no mention of mechanical properties is made anywhere in the work. This undermines the integrity and validity of the study.
- “The friction torque and variable coefficient characteristics mean that the CBFM B is more severely affected by pressure. The continuous notation hole creates confusion for the reader. Uniform characterization notation should be used and this attitude should not be abandoned until the end of the study.
- “Once the applied pressure becomes constant, the roughness contact dominates the bonding process, which not only prevents the formation of oil films, but also accelerates the cracking of the oil films.” Another reason for this cracking is the micro-level vibration, so the vibrations created by the samples should also be shared.
- “Furthermore, the less porous contact surface leads to a thin oil film of B, which allows the contact torque to predominate in the shortest possible time.” Since no density and porosity analysis and determination have been made to support this hypothesis, it cannot go beyond this claim. Therefore, if the authors have done density and porosity analysis, they should be shared in the study.
- Here, the authors persistently attribute the decrease in porosity for samples with SCN addition, and therefore the increase in heat and friction torque, to this. As stated above, this finding needs proof, since no analysis and results to prove this were included in the study.
- It is known from the basic principles of tribology that; The mechanical properties of the couples also have important effects on tribo interactions. In addition, the increase in temperature and friction torque due to tribo contact are also related to hardness and tensile strength, and this cannot be ignored.
- In addition, the authors added an uncanny amount of SCN added to test samples? Why do they not talk about the strong lattice structures of SCN nano-ceramic particles and the release of high bond energies.
- “As the applied pressure increases, the generated heat dominates the clutch temperature. The higher average COF of B eventually results in a higher rate of temperature rise. At the same time, the higher average COF gives the B greater frictional force resulting in a shorter engagement time. As a result, SCN does indeed improve the COF of friction materials.” What is the ideal limit for the COF value. It has been reported in the literature that extremely high material hardness will also cause negative effects (vibration) for engagement and gearbox (doi:10.1016/j.triboint.2016.03.00). The authors were content with the addition of a single type of SCN, the amount of which we have not yet learned. What is the optimum SCN ratio that provides the ideal COF value?
3.1.2. ATF temperature
- “As the applied pressure increases, the generated heat dominates the clutch temperature. The higher average COF of B eventually results in a higher rate of temperature rise. At the same time, the higher average COF gives the B greater frictional force resulting in a shorter engagement time. As a result, SCN does indeed improve the COF of friction materials.” What is the ideal limit for the COF value. It has been reported in the literature that extremely high material hardness will also cause negative effects (vibration) for engagement and gearbox (doi:10.1016/j.triboint.2016.03.00). The authors were content with the addition of a single type of SCN, the amount of which we have not yet learned. What is the optimum SCN ratio that provides the ideal COF value?
- “And B's these are 19.46, 20.12, 18.6275 and 19.7059. Since B has higher temperature rise rates than A, changes in ATF temperature will cause B to drastically increase temperature. It can be noticed that the thermal stability of B is much worse, leading to acute wear at high temperatures.” Since no information is shared about the elemental composition of the SCN added test samples, it can be thought that it would be appropriate to make the following inference only. Thermal conductivity of composite test specimens with SCN added due to increased ceramic ratio in the composite resulting from the increasing SCN nanoparticle ratio.
3.1.3. Rotating speed
- “The difference in microscopic morphology leads to a different friction torque trend at low speeds.” Authors should clearly state what they mean by this sentence (Porosity? Surface roughness?).
- “A's frictional stability is much better as the porous surface of A can store more oil to form a thicker oil film during the clutch operation.” Do the authors mean roughness by porous structure? Since this situation is not clarified by the authors throughout the study, it creates confusion in the mind of the reader.
3.2. Wear characteristics
3.2.1. Thickness variations
- “Average thicknesses of the initial and tested discs are shown in Table 2 and Figure 10(c).” The first thickness measurements and post-test thickness measurement results of the samples used in the Tests of this sentence are shown in Table 2 and Figure 10(c). it should be in shape.
- “P2 is smaller in thickness than the others, and such a phenomenon is even more pronounced in B.” It was observed that the P2 friction discs of both types were worn more than the others after the wear test. This situation has been observed to be greater in the P2 disc with SCN addition compared to its competitor.” It should be in shape.
- Authors should avoid unnecessary waste of words and adopt the way of expressing situations and facts with clear, concise and carefully chosen sentences that will make it easier for the reader to understand.
- “Fig. 10. Initial and tested thicknesses of A A-I, A-P1, A-P2 and A-P3 (a); Initial and tested thicknesses of B B-I, B-P1, B-P2 and B-P3 (b); average thickness of the discs(c) and amount of wear(d)” Figure text is too complex and content is not clear. Using a general expression such as "pre-test and post-test measured thickness values" rather than "tested thicknesses" will clear up the confusion.
3.2.2. Micro-morphology variations
- It is appropriate to present Figure 11 and Figure 12 under this heading. In scientific studies, it is a common understanding to share figures and tables under the aforementioned sentence.
- “Fig. 12. Tested surface morphologies of B (CBFM with SCN) (a)-(c).” The use of double notation is also available here. Because the authors themselves experienced the confusion caused by the use of double notation, they resorted to an additional explanation (CBFM with SCN). From the reader's point of view, it is necessary to clear up this confusion and confusion encountered throughout the article.
- “Fig. 12. Tested surface morphologies of B (CBFM with SCN) (a)-(c).” Figure caption “Fig. 12. Post-test white light interferometric profilometry (a)-(c) of test sample with added SCN.” It would be more appropriate to arrange it in the form. It would be appropriate to make the same adjustments in Figure 11.
- For the white light interferometric profilometries presented in Figures 11 and 12, reference magnifications such as “100 times over an evaluation length of (…µm)” should be specified.
- It would be a more accurate approach to share the Rz value, which is accepted as the reference value for the surface roughness in systems operating as tribo pairs, from the 2D line image obtained from the white light interferometric profilometries presented in Figures 11 and 12.
- An appropriate approach would be to share superimposed (scanned over an area of … mm x … mm) unfiltered stitched 3D height data (Sq) generated from white light interferometric profilometry.
- It would be more appropriate for the authors to use micrographs taken in SEM for the surface characterization of the friction test specimens and counter-reaction discs formed by the addition of SCN particles. Because SEM analysis provides quantitative information about the chemistry, size and amount of the phases and particles present, while morphological analysis provides information about the physical relationships of the size, crystallinity and juxtaposition of the existing phases. The combination of higher magnification, greater depth of focus, greater resolution, and ease of sample observation make SEM one of the most used tools in experimental material and surface characterization. Considering the contribution of these advantages to the study, the authors are recommended to perform their analysis of wear mechanisms and surface characterization with SEM micrographs.
- Positioning Figure 3 in this section will be the right approach for readers to make an easy comparison.
- “However, there are many pores much larger than B's scattered across the contact surfaces of A.” During the relative movements of the tribo couples, these gaps may have been formed by the pulling of micro-scale particles in the CBFM test samples at high speeds. Again, during the test, delamination or layer delamination may occur due to pressure and micro-scale high pressure (Hertz pressure distributions and deformations caused by these pressures) and thermal shocks. White light interferometric profilometry
- One of the reasons why this situation is not observed in the test samples with SCN addition can be considered as the increased mechanical properties.
- “Also, the Sa and Sq morphology parameters of discs with B are much smaller than those of A, which means that the wear process of B is more severe.” Analysis over thickness prevents accurate and clear analysis. Authors are advised to calculate mass loss or specific mass loss and include it in the article. This will allow an easier comparative clear analysis to be made.
- “Also, the Sa and Sq morphology parameters of discs with B are much smaller than those of A, which means that the wear process of B is more severe.” Figure 1, Table 2, Figure 10, and Table3, Figure 11-12, when evaluated together, require evidence even if the assessment made is correct, since the authors did not share information about the Sa and Sq morphology parameters of the discs before any test for friction test specimens.
conclusions
- “The friction-wear properties of CBFM with SCN additives were investigated considering the applied pressure, rotational speed and ATF temperature. Meanwhile, friction characteristics were evaluated by friction torque and temperature through the SAE#2 clutch bench test; The thicknesses and microscopic morphologies of the tested discs were obtained to show the wear status.” Since the statement is repetitive information, this statement should be removed from here. This section directly states “The results from this study are as follows.” It should start with the statement.
- “CBFM with SCN exhibits higher friction torque and shorter engagement time under different operating conditions, suggesting that SCN additives may help increase the COF of CBFM.” This statement should not contain probabilities. Because the authors showed with graphs that the contribution of SCN in the study increased the COF of CBFM of SAE#2 clutch bench test results.
- The notation “CPBM” is an expression that has never been used before. The authors are advised to correct this notation as they may be mistakenly used.
- “As SCN changes the compactness and microscopic morphology of CBFM, the initial microscopic morphology of SCN and CBFM is flattened with a less porous area. Thus, the SCN additive causes more severe wear on the friction components, especially for the P2 disc.” The authors should use SEM and similar microstructural analysis tools and methods for these determinations. Therefore, in the study, the authors do not perform microstructure analysis from surface characterization tools, white light interferometric profilometry.
- “Thus, the SCN additive causes more severe wear on the friction components, especially for the P2 disc.” It may not be accurate for the authors to conclude that the SCN additive causes severe wear. At this point, to the authors; It is recommended to report the negativity of this ratio, emphasizing the ratio used in the manufacture of SCN-added test specimens.
GENERAL EVALUATION:
Authors are required to rearrange the article by paying attention to the scientific text fiction in the article. The results of the test and analysis of the article and the conclusions of the authors contain some fundamental contradictions. Authors should pay attention to the epistemological origins of the concepts and use the concepts correctly where appropriate and necessary. The work contains significant errors and authors are encouraged to resubmit the work with a major revision.
Since the recommended studies are up-to-date, it would be appropriate to include them in the article.
https://doi.org/10.3390/ma14154217
Reviewer 3 Report
- The abstract looks good. Please include significance numerical results
- For the introduction section, please add more reference and briefly explain them.
- In the last paragraph of the introduction, it should be expressed the novelty of the study, the differences from the past in detail.
- Please provide more information about SAE#2 bench and support it with figure(s). Also, no reference is made to the test device.
- What is the production process of test samples? Please mention in the relevant section?
- How CBFM A and CBFM B material contents were determined? What are the SCN contribution rates in CBFM B? Please explain in the relevant section.
- “…the inner and outer radii(mm), respectively.” Please review this sentence
- Results and discussion and conclusion parts are inadequate according to citation and analyze in detail. There should be the importance of the study in detail, comparison results with other approaches in literature, the success of the prediction and computational results.
- Improve the results and discussion and conclusion parts.
- Please fix the typographical and eventual language problems in paper. (e.g. , and)
- The paper is well-organized yet there is a reference problem. First, your reference list contains no paper from “Materials” journal. If your work is convenient for this journal’s context then there are many references from this journal. Secondly, cited sources should be primary ones. Namely, indexed area shows the power of a paper and directly your paper’s reliability. Please make regulations in this direction.
*** Authors must consider them properly before submitting the revised manuscript. A point-by-point reply is required when the revised files are submitted.
***To improve the quality of the paper these papers can be useful:
1) https://doi.org/10.3390/ma14185145
2) https://doi.org/10.3103/S1067821221010077

Reviewer 4 Report
The paper entitled; " Effect of silicon carbide nanoparticles on the friction-wear properties of copper-based friction discs” is an interesting article that contains exciting experimental and applicable results. Therefore, it can be accepted in your valuable journal after doing some major revisions which can be summarized as the followings:
- Abstract must consist of the purpose and methodology along with significant findings, which is lacking in the current article. need to be changed.
- What is the novelty for doing this research experimentation?
- The objective of the research must be written in a way that shows the new and its difference from previous research.
- In the manuscript, the authors relied on Cu and improved by adding SiC nanoparticles. There should be a paragraph in the introduction about the properties and importance of Cu based and another about SiC, advantages and reason for choosing it as reinforcement. The following references can help you:
- Essam B. Moustafa, Mohammed A. Taha, Evaluation of the microstructure, thermal and mechanical properties of Cu/SiC nanocomposites fabricated by mechanical alloying, International Journal of Minerals, Metallurgy and Materials, 28(3) 92021) 475.
- Some of mistakes need to be carefully reviewed.
For example page 5: the Tmax and Tmin of A are 66N·m and 34N·m, and these of B are 88N·m and 53N·m. should be corrected to , the Tmax and Tmin of A are 66 and 34N·m, and these of B are 88 and 53N·m..
Page 8 : the Tmaxs of A under pre-set pressures are 460N·m, 460N·m, 420N·m and 377N·m, respectively, and these of B are 430N·m, 451N·m, 395N·m and 374N·m. should be corrected to the Tmaxs of A under pre-set pressures are 460, 460, 420 and 377N·m, respectively, and these of B are 430, 451, 395 and 374N·m..
And so on.
- The quality of most figures should be improved.
- The language must be carefully reviewed.

Round 2
Reviewer 1 Report
The article is acceptable as it is.
Author Response
Thank you very much for your acceptance.
Your valuable comments have greatly improved my article!
Reviewer 2 Report
I attached my report.

Reviewer 3 Report
- A figure related to the production of test samples should be attached to the article.
- English spelling rules need to be revised. The article cited by the authors is not exemplary. Authors should do proofreading for the article and upload a document about it.
-
https://doi.org/10.1016/j.jmrt.2021.11.114
Authors should expand the introduction with the articles given above. (The articles whose doi are given should be mentioned.)
After these corrections, I think that the article can be published. I regret to inform you that this article will not be an adequate article for this journal without these corrections.
Reviewer 4 Report
The authors carefully made all necessary adjustments. Therefore, I recommend that you accepted the manuscript.
Author Response
Thank you very much for your acceptance.
Your valuable comments have greatly improved the article!